# Description or Truth? A Typology of New Testament Theology

**Seth Heringer**

Department of Bible and Theology, Toccoa Falls College, Toccoa Falls, GA 30598, USA; sheringer@tfc.edu

**Abstract:** This essay develops a typology to divide the overcrowded disciplinary space of New Testament theology into eight approaches based on subject matter. After describing and analyzing the approaches, it argues that descriptive New Testament theology becomes unworkable due to internal tensions. Next, it evaluates a recent proposal by Robert Morgan for "implicit" theological interpretation in New Testament theology. After finding Morgan's approach to insufficiently distinguish itself from a descriptive history-of-religions account, it argues that the future of New Testament theology must consist in a move away from description and toward a search for truth. It encourages Christians to read the New Testament in ways consistent with their own beliefs. The essay concludes by arguing that the future of New Testament theology is one of self-sacrifice in order that something better may appear.

**Keywords:** New Testament theology; biblical theology; theological interpretation; typology; historical criticism; history of religions; description

## 1. Introduction

Claims to the labels "New Testament theology" (NTT) and "Biblical theology" (BT) are widespread in contemporary biblical studies.[1] Publishers have encouraged this trend in titles ranging from *Ice Axes for Frozen Seas: A Biblical Theology of Provocation* to *Anthropology and New Testament Theology* to *Many Roads Lead Eastward: Overtures to Catholic Biblical Theology* (Brueggemann 2014; Maston and Reynolds 2018; Miller 2016). A comparison of similarly-titled books shows that they contain extensive methodological diversity, indicating that BT and NTT do not designate strict disciplinary boundaries. Rather, these phrases serve as aspirational and promotional signals that designate certain books as faithfully representing the content of the Bible. In other words, these phrases function as corporate slogans. Disney is where dreams come true; Gillette is the best a man can get; NTT is where accurate descriptions of the content of the New Testament can be found. Or differently, claims to be doing NTT or BT have become a battlefield where the winner collects the spoils of credibility. Those who occupy the center defend the border with disciplinary skirmishes, claiming that the invaders have no right to this ground and the respectability that comes with it. Those inside the borders reap academic prestige whereas those outside are left in ignominy. Professional advancement may be at stake if a scholar's work cannot claim the mantle of NTT or BT.

Given this context, what future does NTT have aside from being a contested naming scheme? Part of this answer comes from looking at its past.[2] But a discussion of its future must ask what the field should be and not just what it currently is. This essay will take up that prescriptive and aspirational task with the following plan.[3] First, I will lessen the importance of the term "New Testament theology" by developing a typology that divides an overcrowded disciplinary space into better defined approaches based on subject matter. Second, I will use this new methodological clarity to argue against descriptive approaches to NTT and for approaches that push beyond description by adjudicating the truth claims of the New Testament. Third, I propose that Christians use an intentionally confessional approach, what I call Scriptural theology, because it encourages Christians to read in alignment with the claims of the New Testament and consistent with their own beliefs.

## 2. A New Typology of NTT

Scholars could decrease the intensity of the debates over NTT by better delimiting the field according to *what* is being studied. All that is called NTT does not study the New Testament nor is it theological.[4] One possible approach to do this would be to create a typology that groups examples of NTT according to similarities.[5] I have chosen a different approach and created a typology formed from the answers to the following three cascading questions.

1.　Does the NTT study the text of the New Testament or the history and context behind it (text or history)?
2.　Does it stop with a description of the authors' claims or push further to adjudicate whether those claims are true (description or reality)?
3.　Is the subject matter applicable to modern readers (neutral or prescriptive)?

The varying answers to these three questions create the eight heuristic approaches displayed in Figure 1.

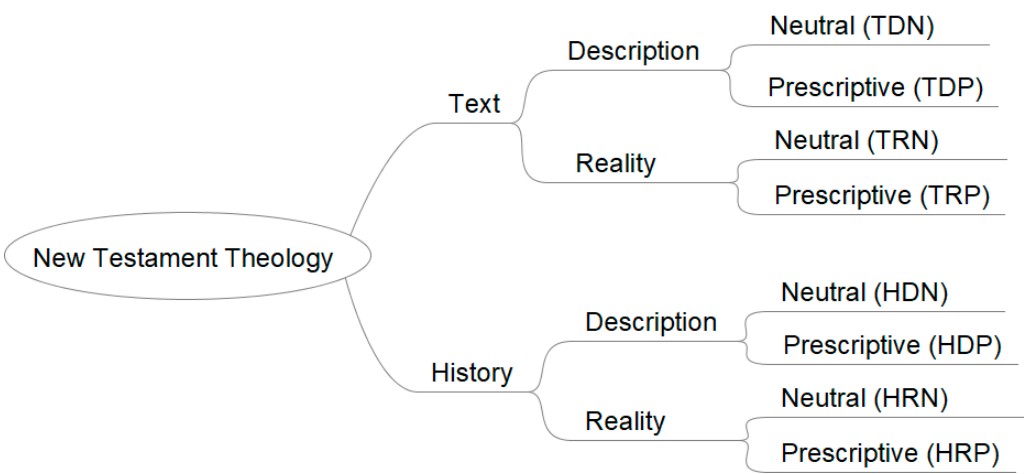

**Figure 1.** Typology of NTT subject matter.

The first question begins by distinguishing between textual and historical approaches to NTT. To clear up possible ambiguity, seeking the mind of the author or the beliefs of a community would be to pursue the history behind the text and not the text itself. Some may object and argue against a strong division between the study of a text and its context, stating that the meaning of a text can only be discerned by knowing the mind of the author and how his community used words. It is true that some level of historical knowledge is needed both to read Greek and know the semantic range of words. The distinction here, however, is about focus, effort, and goals. If the focus and primary effort is spent reconstructing the past as it really was, and if the New Testament is seen as a source to get at the minds of its authors, then the history behind the text is the goal of the work. If, alternatively, the focus and primary effort is to understand the grammatical sense of the text and use history as a secondary tool in order to understand its language use, then the text remains the primary endpoint.[6]

The second question assumes that both branches of the first step in the typology lead to texts, authors, or communities making claims about life and reality. Once these claims are identified, it asks if the biblical scholar is satisfied to let her work only describe ancient beliefs, or whether she wants to push further and investigate if those beliefs speak truth about reality.

The third question asks whether the claims of the texts, authors, or communities make prescriptive claims on the reader. Can these claims cross the historical ditch between the past and present? To clarify, this question does not ask if texts, authors, and communities make ethical claims, for surely, they do, but whether those claims have any pull on the present.

Although structuring the typology according to these questions allows for clearer logical distinctions among the approaches and a stronger analysis of the internal coherence of each, it makes giving clear examples difficult because the typology was not formed by the contours of existing New Testament theologies. Most examples of NTT cut across these eight approaches, mixing them to various extents. Therefore, when examples are given, they do not designate a strict identity but an orientation toward an approach.[7] Additionally, an important distinction must be made between subject matter and methodology. In this typology, a biblical scholar could use any method and still pursue the text (and not history) as the subject matter. Most likely, she will choose a literary method, but is not limited to that. She could use every historical and sociological tool available in order to understand the text as best as she can. Some methods, however, will fit better with certain subject matters, but that analysis is beyond the scope of this essay.

When asked in numerical order, these three questions form the eight approaches described below.[8] Some approaches are more hypothetical than actual, which means not all are represented by a clear example of an existing NTT. Some approaches are also internally more coherent than others, but that analysis will be made after describing each.

TDN: A description of the meaning of the New Testament that makes no prescriptive claims on the contemporary world. It resembles pure literary studies, something akin to Shakespeare studies that investigates only the literary meaning of the text. An example would be a NTT that examines Jesus's parables for their grammatical meaning and narrative purpose without considering their ethical import for today. An example is *A Narrative Theology of the New Testament: Exploring the Metanarrative of Exile and Restoration* (Eskola 2015).[9]

TDP: A description of the meaning of the New Testament that makes prescriptive claims on the contemporary world. It would resemble ethically-informed literary studies, which would see *1984* not just as a good work of literature but a warning against authoritarianism and its complete control of information. Similarly, it would look at the grammatical and narrative meaning of Jesus's parables and reflect on their relevance for modern life. Schreier's *New Testament Theology: Magnifying God in Christ* serves as an example (Schreiner 2008).[10]

TRN: A work that adjudicates the truth of New Testament claims while making no ethical entailments for the contemporary world. In other words, it takes the claims of the New Testament seriously and investigates them with whatever methodology the scholar finds appropriate. The results of this investigation remain unconnected to the lives of modern readers.

TRP: A work that adjudicates the truth of New Testament claims and makes prescriptive entailments for the contemporary world based on the results. For biblical interpretation, it would investigate whether the Bible made true claims and how those claims affect various aspects of human life. An example would be the approach Hans Frei sets out in *The Eclipse of Biblical Narrative: A Study in Eighteenth and Nineteenth Century Hermeneutics* (Frei 1974).[11]

HDN: A description of the New Testament authors' beliefs and the historical context surrounding them that makes no ethical claims for the contemporary world. An example from historical studies would be a book that described the theological beliefs of the ancient Greeks without making any claims as to their truth. Did Hera hate Hercules? This approach is silent on their reality and rancor. An example is the method John J. Collins argues for in *Encounters with Biblical Theology* (Collins 2005).[12]

HDP: A description of the New Testament authors' beliefs and the historical context surrounding them that makes ethical claims for the contemporary world. There is no parallel in historical studies since it would require investing ancient beliefs with a prescriptive authority even though the author made no attempt to validate those beliefs as true. The mere words on the page are authority enough. Thomas R. Hatina

proposes a similar method in his *New Testament Theology and Its Quest for Relevance* (Hatina 2013).[13]

HRN:A work that adjudicates the truth of the New Testament authors' beliefs and the historical context surrounding them using scholarly tools while making no ethical claims for the modern world.[14] To return to claims about the ancient Greek gods, here, the reality of those beliefs would be investigated while keeping the results disconnected from the modern world. NTT would do the same regarding claims about Jesus walking on water or healing the sick. An example is *A New Testament Theology* (Blomberg 2018).[15]

HRP:A work that adjudicates the truth of the New Testament authors' beliefs and the historical context surrounding them using scholarly tools while making normative claims for the modern world. Did Jesus really rise from the dead? The answer to that question, it says, should determine how you live your life. An example would be the approach described in *Beyond New Testament Theology: A Story and a Programme* (Räisänen 2000).[16]

### 3. Analysis of the Typology's Approaches

Some of the eight approaches are worse options for the future of NTT than others due to internal tensions and logical incoherence. Because these approaches have been created by the answers to a series of questions, the following arguments are not directed against any particular NTT but logical constructs. The first problematic approaches are TDP and HDP due to description (D) serving as an inadequate basis for prescription (P). Why would a modern person submit to the beliefs of an ancient text, community, or person without investigating whether they are true? On the one hand there is a possibility for weak prescription akin to what is offered by fables and parables. They inspire, warn, or give hope. They teach lessons about the world and describe human relationships. They do so, not because of an inherent authority, but because the reader recognizes that somehow, perhaps metaphorically or analogically, they shine light on reality. If this weak basis is used for prescription, then the New Testament is read like any other book of stories or fables that surfaces the internal wisdom of the reader, and a reason to study this book over all others slips away. On the other hand, religious authority can confer a stronger basis for prescription by guaranteeing truth. This alternative foundation for truth is limited only to confessional contexts. So TDP and HDP either treat the New Testament like any other book or limit themselves to strongly confessional readers.

The question of truth offers an even broader challenge to all four descriptive approaches to NTT. This challenge is best framed with a question: is it possible for descriptive accounts to remain descriptive no matter the content or who is speaking? To be sure, it is possible to offer a neutral description of topics that do not affect the reader such as a book about the history of European ferns. Moreover, a descriptive approach becomes attractive when an author does not want to make conclusions about a topic that may be offensive or lead to unwelcome results. As an example, it is much easier to take a descriptive approach to the claims of indigenous religions rather than question their truth.

There are two places, however, where cracks appear in the wall that descriptive accounts construct against the questions of truth and reality. The first crack appears when a text, author, or community makes claims that would, if true, directly affect the one describing those claims. For example, imagine a doctor looking at a patient and exclaiming, "You are having a heart attack! You will die if we do not get you into surgery immediately." The patient could offer a descriptive account of this encounter and say to his friend, "The doctor said that if I don't have immediate surgery I will die." This is the proper response for a descriptive approach for it describes the beliefs of the doctor without forcing the man to make a decision about his health. However, this is an absurd response. Surely, any right-thinking person is forced to make a decision and not remain in description. The nature of the claim pushes on him in such a way that forces an investigation of its truthfulness and spurs actions based on the result. This is not a book about ferns; instead, this is a

description that requires action. If an official approaches and says, "Your lottery ticket won the jackpot", the wise response would be to investigate this claim and act upon it. To remain descriptive here would be to preclude any action that could bring joy. Thus, statements of this sort pressure the hearer to move from description to reality.[17]

The second place cracks appear in the descriptive wall is when the description is of the claims or beliefs of a particular sort of person. For example, if a wife tells her husband, "I would like to spend some time together this weekend", the husband could take a descriptive approach and describe to his friend how his wife would like to spend time with him. However, a wiser plan would be for the husband to first decide if his wife is speaking truthfully and then act upon it. If he decides she is speaking earnestly, he would rightly hear in her words a prescription. Particular statements about the world by special people, such as loved ones, puts pressure on the hearer to move beyond description to prescription. Or consider an emperor who says to his entourage, "I am hungry." The hearers may stick with describing that state to each other, but that reaction may quickly lead to fatal consequences. There is no command in the emperor's language, but the wise person realizes who is speaking, and if he believes the emperor, will rush to satiate him.

When NTT tries to remain descriptive, both of these cracks expand. Many claims in the New Testament would affect the person describing them if true and are spoken by somebody special. An example of this comes when Jesus recounts the coming of the Human One to judge the nations and separate them into sheep and goats. Jesus identifies the sheep as those who fed the hungry, gave water to the thirsty, welcomed the stranger, clothed the naked, and visited those who were sick or in prison; the goats being those who failed to do these things. He then speaks judgment on the goats when he says they "will go away into eternal punishment" whereas the righteous ones "will go into eternal life" (Matt 25:31–46).[18] A descriptive approach would either describe the grammatical meaning of this text or the minds of the authors or community that created it. Some would even try to find prescriptive meaning for the modern world in these words. But using a descriptive approach to this text is as unwise as a using a descriptive approach to a warning of an impending heart attack. Both may have fatal results. The very claim itself, that eternal punishment or eternal life is at stake, requires that the truth of the claim be investigated and acted upon. Not only are there consequences involved, but the claim is being made by a particular sort of person: a person the New Testament describes as the Son of God, creator, and redeemer of the world. Two questions must now be investigated: is Jesus who the New Testament says he is, and if so, is this coming judgment real?[19]

A defender of descriptive NTT might reply that although some claims require moving beyond description, that work is done by the readers of NTT, not the authors. The New Testament theologian merely describes the best understanding of the text, author, or community and the reader is left to make the appropriate investigations into their truth claims. This reply fails for three reasons. First, if the claims of the New Testament require a response, then the author of the NTT should have already completed the task of assessing the truthfulness of the claims. If she has assessed that Jesus is the Son of God and the coming judgment is real, or vice versa, then how could she keep this from her readers? Second, many texts such as the Great Commission (Matthew 28:18–20) speak about the need for proclaiming the message of the gospel and therefore push the author to do so in her work. Additionally, Jesus's warning that "Whoever is ashamed of me and my words, the Human One will be ashamed of that person when he comes in his glory and in the glory of the Father and of the holy angels" (Luke 9:26) shows that the fear of losing scholarly status due to proclaiming her own beliefs cannot serve as a reason to retreat to description. Third, there are claims that are so morally problematic that they cannot be merely described neutrally. As an odious example, consider this pro-enslavement writer from the antebellum South describing why southern enslavers will not mistreat enslaved black men and women: "His [enslaver's] interest in the life and health of his slave obviates the necessity of any particular supervision of the subject by the public authorities. No better security has ever yet been devised by man, for the safety of man, and the proper observance of humane laws



by the citizen, than that which the Southern slaveholder offers, in the continual presence of his leading interests" (Simms 1853, p. 228). A purely descriptive retelling of this account would look at the grammatical range of meaning or do an investigation into the mind of the author and his community. But is it possible to just stop there? Must we not release the fire that wells up in our belly against the deceit of those words? Is not this topic so significant as to require moving beyond description? There are other such topics where a descriptive approach is morally problematic. Claims about God's existence and character determine the moral order of the universe. If these claims are true, then the author cannot remain silent about them because they are too weighty. Their nature presses on the author to seek and speak truth about them.

Descriptive approaches are not the only ones with problems, for TRN and HRN both make judgments about truth and reality (R) while jettisoning any discussion of its effects on the modern world (N). On the one hand, it is possible to remain neutral about the truth of unimportant things. To return to our example of a book on European ferns, the truth of this book makes no claims on a modern reader. However, it is much harder to remain neutral about truth claims central to life and existence. If I am about to walk through a field and read a red-lettered sign that says, "Warning: Landmines", what is the only reasonable thing to do once I believe that testimony? There is a reality outside of myself that I must take into account. My assessment of the truthfulness of the sign requires a modification to my life. If we expand this idea up to the level of theology, it becomes significantly more implausible to say that the reality of God's existence has no effect on the modern world, my existence, or life. If a NTT assesses that the resurrection is real, does that not change everything? If it assesses it is false, does that not do the same? To say that such claims have no relevance to modern readers is put up false barriers blocking the author from dealing with what is directly in front of her.

Before turning to TRP and HRP in the penultimate section of this essay, it is worth emphasizing that every descriptive approach exhibited problems with internal coherence. Does that finding signal larger problems with descriptive approaches to NTT? Perhaps these problems only appear in the abstract and are avoided in practice? If descriptive NTT is to be saved, it will have to be done by looking at concrete proposals.

## 4. Robert Morgan's Implicit Theological Interpretation

Robert Morgan has recently proposed an HDN approach that he believes is able to unite NTT's scholarly and theological character (Morgan 2016, 2018).[20] I have chosen to focus on this proposal because, unlike most examples of NTT, Morgan spends a significant amount of time defending his methodology.[21] Additionally, Morgan's desire to create an approach that can be used by both secular and confessional scholars alike makes it an especially attractive proposal to investigate to see if descriptive NTT is viable.

His proposal is historical (H) because he thinks that discovering the thoughts of the New Testament authors is the subject matter of NTT. "New Testament theologians", he says, "normally describe and try to explain the biblical authors' ancient understandings of their faith, and they do this in awareness of their own personal and ecclesial standpoints." He expands on this idea by saying that the goal of NTT is not primarily "historical description of the human realities behind these texts" but the "interpretations of the texts themselves, interpretations aiming to communicate what the original authors intended" (pp. 385, 390). Morgan is not dismissing historical events as unimportant but emphasizing the priority of getting to the minds of the authors.[22] Additionally, the above two quotations explicitly claim his approach is descriptive (D) in its goal to recover authorial intent without making judgment as to its veracity. The writer of a NTT should not make such judgments because "New Testament theology, as a largely historical discipline, has attempted to present original meanings and is typically silent about the interpreter's religious interests and theological standpoint" (Morgan 2018, p. 205). This standpoint silence reinforces the descriptive nature of NTT by precluding any interaction of the text and the New Testament theologian's understanding of reality and truth. Morgan's approach is neutral (N) because

it rejects normative language in NTT: "Biblical scholars across the spectrum from Wrede to George Ernest Wright and N. T. Wright describe the ancient writers' religion and their texts' talk of God without themselves regularly making normative theological statements" (Morgan 2016, p. 389). Thus, NTT should avoid making statements about the reality of New Testament claims that could intrude on the lives of contemporary readers. So far, Morgan's approach resembles a descriptive history-of-religions approach to NTT.[23]

Despite the similarity, Morgan criticizes examples of NTT that go too far in their secular orientation. For instance, he believes the history-of-religions approach forsook any theological character by reconstructing the history behind the text absent theological concern. William Wrede, according to Morgan, constructed just such a theology-less approach.[24] Morgan thinks Wrede made a "category mistake" by reflecting a "biblical scholarship whose critical historical achievements had outrun its hermeneutical reflection" (p. 388). Morgan's approach attempts to keep the critical achievements of Wrede's conclusions and methods while improving its hermeneutical reflection by promoting different aims and motivations. These revamped aims and motivations are how he adds theology to a descriptive history-of-religions approach. These theological aims and motivations cannot encroach on the "historical and exegetical tasks", Morgan warns, and "must surely be distinguished from modern theological judgments." In other words, Morgan has a vision of NTT as "theological interpretation of these texts within the constraints of modern scholarship." He wants this mixing of theological interpretation and scholarship to create a NTT that "sounds reasonable to outsiders and insiders alike" (pp. 387–89). It is here, at the intersection of the historical method and theology, that Morgan introduces his solution to the tension between them: implicit theological interpretation.

To review, Morgan is trying to create an approach to NTT that has a theological character while retaining the methodological strengths of a descriptive history-of-religions approach. This goal requires a novel understanding of the word "theological" due to the recurrent tension between speaking openly of God and the accepted rules of the historical method. Rather than "theological" referring to a characteristic in the text of a NTT, such as explicit talk of God, it instead refers to what happens in the mind of the scholar. Such a definition is useful because it allows for almost all biblical scholarship to be labeled "theological." The ubiquity of such a theological mindset, Morgan argues, is shown in that "Most Christian scholars more or less agree with the New Testament about who God is, and about the central significance of Jesus, and some of them have allowed their personal convictions to shine through their scholarly work."[25] Morgan identifies these "some" as doing "explicit theological interpretation" that speaks openly of God (pp. 384–85). Speaking of God explicitly, however, is not the only way for a work to be theological because scholars "may depend on systematic theology in shaping their own theological standpoint and in the application of their conclusions to contemporary Christianity" while doing "their scholarly work without alluding to their own standpoints. As they allow their historical, social-scientific, or rhetorical constructions to speak for them, the theological interpretation going on in their heads can remain implicit in their writing" (p. 392). The author of any NTT, Morgan argues, will have biases and beliefs. That is acceptable. In fact, such beliefs are desired. What is not desired is letting those beliefs become apparent in the work itself. They must remain implicit.

It still needs to be clarified what exactly makes implicit theological interpretation "theological." For Morgan, it is not something in the text, such as a method or even the subject matter; rather, "it has been the aims and assumptions of some interpreters that have made their work theological" (p. 386). Such a mindset shares the same standpoint as the New Testament authors: "Theological interpretation of these canonical texts is undertaken by the relevant rational methods on the assumption (whether hypothetical or genuine—some theological interpreters are more or less agnostic) that what the texts say about God refers to transcendent reality. That means that mutatis mutandis (on account of changing worldviews) theological interpreters share the biblical authors' standpoint in relation to the religious tradition whose meanings as contained in these texts they are

aiming to communicate" (p. 390).[26] Morgan holds that as long as an author believes the New Testament refers to a transcendent reality (and even if this belief is agnostic or hypothetical), then whatever that person writes about the New Testament should be considered "theological." With this approach, the model NTT would be written by a Christian who generally believes the New Testament's claims about transcendent reality, writes descriptively about the authorial intent of the authors, and makes no adjudication of the truth of these claims so as to not let her beliefs appear in the text.

Morgan anticipates the objection that his approach is too timid and self-limiting to the point of "apostasy" (Morgan 2018, p. 214). He says some will see him, similar to Nicodemus, as using the night to hide his conversation with Jesus, using implicit theological interpretation to hide scholars' true beliefs in order to gain "the repute of this world's methods and secular careers" (Morgan 2016, p. 387). Morgan rejects this criticism as misreading his intentions. His goal is not plaudits but persisting conversation between theology and the academic world: "NTT has provided a way of preserving in secular institutions the religious aims of most Bible study, and in religious institutions the secular methods used in other disciplines, making conversations possible across the spectrum of biblical scholarship" (Morgan 2018, p. 208). He believes the price paid in losing explicit God-talk is worth the continued conversations between Christian biblical scholars and the academy.

## 5. Analysis of Morgan's Implicit Theological Interpretation

Does Morgan's approach overcome the internal problems of descriptive NTT identified above? To begin, Morgan does not address the internal issues because descriptive NTT has not identified them a problem. It has not yet wrestled with how to justify stopping at description and not progressing further into questions of truth and prescription. Beyond those concerns there are additional problems specific to Morgan's formulation. The first arises when considering Morgan's motivation to keep NTT "theological." The academic world does not care if NTT retains its theological character, so this move must be directed toward confessional audiences. But why would such audiences care about a term that is not allowed to have any clear influence on the work itself? Two possibilities arise. The first is that the term is retained for the sake of the confessional author, lending meaning or purpose to the work because it is "theological" even if that characteristic is not apparent in the text. The second is that the term signals to confessional audiences that this scholarship can be trusted because it is done by a person who shares their beliefs. Even if the methods and conclusions of the text contradict the beliefs of confessional audiences, they should have no fear because behind the text the author agrees with them on what truly matters. Trust this NTT, the term soothes, for it mixes theology and critical scholarship in a reliable way. In other words, "theological" is being used like "shibboleth"—a marker of trusted group identity.[27]

Second, using "theological" as an adjective in this manner creates a strange precedence. The adjective is not describing a characteristic of the text or a methodological approach, for such a possibility is ruled out by the nature of implicit theological interpretation. It describes the mindset of the author. This position requires an assessment of every author's mindset before she could be grouped in this "theological" project because an analysis of a text is unable to surface implicit theological beliefs. If this investigation discovered the author was an atheist, she would have her project labeled "atheistic interpretation" and removed from the "theological" group. However, this is the very situation Morgan is trying to avoid in his attempt to open lines of communication between NTT and academic methodologies.

Third, distinguishing between "theological" and non-theological texts confuses readers when there is no difference in the texts themselves. Imagine two New Testament theologies sitting on table in a book store. A woman sits behind the table, and when a patron approaches, she informs him that if he can guess which text is theological, he can have it for free. He reads them both and determines they share the same method and conclusions. He tells here there is nothing in either to distinguish it as theological and therefore the task

is impossible. She replies that he has the wrong understanding of the term "theological", for it refers to the mindset of the author, not something that is found in the work itself. He is thinking of explicit theology; her, implicit. The patron would most likely stomp away having wasted his time trying to discern the impossible.

Fourth, the property "theological" cannot be assumed to transfer from the author to the text. If the theological character of the text is weak enough to be unseen, is it still worth calling "theological?" If the rules of scholarship preclude speaking about God or thinking of God as active in the world, then in what sense does the theological mindset of the author transfer to the NTT? Imagine a scenario similar to the one above, but now with two cookbooks on this same table. One cookbook is written by a world-class chef; the other, by a skilled home cook. Both cookbooks were written according to a strict set of publishing rules that required all recipes to be simple enough for an unskilled reader to complete in 20 min using only six ingredients. The chef will have to set aside much of her skill and passion in order to comply with these rules. Both books will have similar recipes and techniques because the chef has been hindered from displaying her skill. The publishing rules will prevent the "world-class" character of the chef from transferring to the pages. Similarly, if "biblical scholar" replaces "chef" and "theological" replaces "world-class" in this scenario, it is clear that a method contrary to a theological mindset can greatly hinder the transfer any theological character.

Fifth, the cost of hiding an author's religious beliefs is not worth the value of broad conversations with the academic world. Under Morgan's approach, NTT authors are not able to write freely, openly, or passionately about what they believe to be true about the world. All are muzzled by the rules of historical method. Moreover, the approach forces authors into deception, pretending to be a neutral observers to claims that deal with the core of their identity, eternal hope, and ethical world. All of these must be pushed down and confined in order to converse with a discipline and method that reject much of what they hold dear. This price is steep indeed.

Morgan's approach to NTT has not vindicated descriptive accounts of NTT. In addition to issues of internal coherence, descriptive accounts struggle to justify the "theological" character of NTT in any meaningful sense. If descriptive approaches fail as both logical constructs and in practice, the future of NTT must lie elsewhere.

### 6. The Future of NTT: Seeking and Speaking Truth

Returning to TRP and HRP, both approaches seek truth and speak about it boldly. TRP seeks the textual meaning of the New Testament, tests the truth of its claims with a chosen method, and speaks about the results as having contemporary significance. HRP does the same, except it replaces investigating the text with investigating authorial intent or the history behind the text. Rather than focusing on the differences between these two approaches, I will instead examine two broad understandings of history and hermeneutics that could be used by either TRP or HRP.[28] The first understanding views the New Testament with a hermeneutic of suspicion and uses the historical method; the second, a hermeneutic of trust uses a specifically Christian epistemology.

The first understanding continues a history-of-religions approach to NTT by using the best academic methods available to reconstruct the historical reality behind the New Testament. This approach reads with a hermeneutic of suspicion, always doubting the claims of the New Testament until they can be verified. Stephen L. Young has recently expanded on this approach and given it an ideological and intersectional character. Young does not discuss NTT methodology specifically, but offers a broad criticism of New Testament studies by saying that it does not go far enough in its suspicion, gives too much credence to the text, and participates in protectionism. He defines protectionism as "the collapsing of inquiry into description such that the perspectives of those being studied are privileged in scholarly analysis. Insider perspectives are thus protected, if you will, from interrogation." Specifically, scholars "take these texts at face value" and let the texts "become normative for our scholarship rather than additional materials for us to historicize"

(Young 2020, pp. 329–30). This protectionism merely shields white, male scholars who already have privilege and excludes women and other disadvantaged groups. Instead of this retrograde approach, Young wants New Testament studies to move away from description to explanation and use all available methodologies to explain the experiences behind the text.[29] Etic sociological investigation is a particular tool Young references, but he does not limit explanation to merely one methodology and is open to historical investigation as long as it is not done with a protectionist bent. This understanding treats the New Testament as any other book: it possesses no privileged status, requires no special method by which it must be read, and gives no special access to truth. Christian scholars who take a TRP or HRP approach and desire, similar to Morgan, to follow the accepted methods of the scholarly world in order to enter into broad conversations, will be forced into some version of this understanding.[30] However, for them, there is a more excellent way.

The second understanding is better suited to confessing Christians because it rejects the bonds imposed by the rules of critical scholarship. As we saw above with the analogy of the two cookbooks, those rules are not freeing but stifling; they do not open the horizons of thought but narrow them. This understanding privileges the perspective of the text and speaks explicitly of God. Douglas Campbell has argued for such an approach by saying that there is "only one way to do New Testament Theology" because "we must *begin* with God-talk, so with theology, and, moreover, with God-talk undertaken in a certain way" (Campbell 2021, p. 2). For Campbell, the basis of confident God-talk, and therefore NTT, is the revelation given in Jesus Christ. What is revealed "is indeed the truth—the truth above all other truths. It is to be relied upon where all others fail, and to be acknowledged and maintained under any circumstances" (p. 4). Because the revelation of God in Jesus Christ is true, all other foundations for God-talk are false and enter an infinite regress that searches ever lower for firmer footing. Moreover, Campbell argues that using any other foundation is disobedient because it rejects the foundation given in revelation. This understanding is far removed from a hermeneutic of suspicion because its trust in the revelation of Jesus Christ serves as an explicitly Christian epistemology. The influence of revelation does not stop with epistemology, however, but proceeds to affect the character of those who accept it. This character is shaped in a formative community grounded in experiencing the presence of Christ together. Christian formation produces the virtues of openness to dialogue and humility that will shape any NTT written by members of this community. In summary, Campbell proposes a vision of NTT rooted in the conviction that the revelation given in Jesus Christ is true and serves as the foundation for any speech about God and the church's communal life together.

Campbell has helpfully surfaced an idea that has only remained in the background to this point: the character, history, and worldview of the person writing a NTT should and will strongly shape it.[31] On the one hand, this is an ancient view. Gregory of Nazianzus taught that writing theology "is not for all people, but only for those who have been tested and have found a sound footing in study, and, more importantly, have undergone, or at the very least are undergoing purification of body and soul" (Gregory of Nazianzus 2002, p. 27). He warns that doing theology without this purification is as dangerous as handling holy objects unworthily, thereby running the risk of severe consequences. Writers of theology must be above reproach ethically and participate in a broad range of pious actions from hospitality to singing psalms to fasting. Moving to a physical analogy, Gregory describes the self-formative task of Christian theologians as being like that of sculptors who need "to look at ourselves and to smooth the theologian in us, like a statue, into beauty" (p. 30). This inner work is done so that the theologian is not temped by pride or passions to think wrongly of God and therefore misrepresent God to the world.

On the other hand, recognizing how the identity of the writer shapes a NTT is also a modern view. Joel Green uses cognitive studies to show that what a person sees in a text depends upon the type of person she is. Green argues that gaps exist in any text and the human mind fills those gaps according to past experiences: "We interpret the present and visualize

the future according to past patterns, generally applying old paradigms in new contexts." Thus, a NTT author will read the gaps in the text according to "conceptual schemes or imaginative structures" by which she understands the world (Green 2016, p. 447). If a person reads the New Testament with a scheme of naturalism, then she will fill in textual gaps with naturalistic explanations. A Christian, however, will fill those gaps with explanations based on a history and conceptual scheme that see God as active and working in the world.

Brevard Childs offers an example of this principle in practice by arguing that having more knowledge about the history behind a text does not necessarily make one a better reader of the Bible. Childs begins by challenging a common assumption of the historical-critical method that "If we could know more about Israel's customs and habits, the stories would automatically become clearer." The problem with this assumption, Childs argues, is that it draws the attention of the interpreter to the wrong place. The story quickly shifts out of focus as "elements which are in the background suddenly are moved to the foreground" (Childs 1980, p. 129). To demonstrate this claim, Childs explores two historical-critical interpretations of the story of Elijah and the prophets of Baal in 1 Kings 18. The first reading uses the vast amount of historical knowledge scholarship has produced on Baal to interpret the story as being about the transfer of Baal's mythological power over fire and water to Yahweh. The second reading focuses on the sacrificial bulls as symbols of the Canaanite fertility cult. Childs argues that both of these interpretations let historical knowledge outweigh the text itself and cause them to miss the text's own pacing and emphasis. A better approach is to assume the text purposefully guides the reader's attention. The text gives little emphasis to Baal or bulls other than that Baal is to be mocked and bulls are to be sacrificed. Instead, the text lingers on Elijah's confidence, Yahweh's altar, the profligate wasting of water during a drought, and God's fire from Heaven. Child's investigation of different interpretations of 1 Kings 18 shows that Campbell, Gregory of Nazianzus, and Green are right: the person looking at the text and the methods and interests by which she reads will profoundly affect what is seen and therefore how she writes a NTT.

## 7. Conclusions

For Christians, the future of NTT cannot be one where they abandon their convictions, read the text according to a hostile methodology, and reach conclusions opposed to their core beliefs. Instead, a future approach to NTT must encourage Christians to speak about what they know to be true with passion and without obfuscation or deception. It must allow Christians to be Christian.

This essay started with the goal of lessening the importance of the term "New Testament theology" by clarifying its disciplinary boundaries. With that goal in mind, the eight approaches created by the typology above can be narrowed into three groups that will define the future of NTT. The first group comprises the descriptive approaches to NTT (TDN, TDP, HDN, HDP). Because this group resembles much of what has been called NTT, I propose it retains that label. If this essay's criticisms against descriptive approaches are persuasive, however, this group's influence will wane. The second and third groups are the two understandings of history and hermeneutics used by the TRP and HRP approaches discussed above.[32] The first understanding (Group 2) that uses a hermeneutic of suspicion and the historical method will retain the name it has taken in the past: "history-of-religions." Although Young has shown that this group is not beholden only to the historical-method, the name can still serve as an umbrella term. The second understanding (Group 3) is comprised of confessional Christians who use a hermeneutic of trust and a specifically Christian epistemology. I propose calling the work of this third group Scriptural theology because it is written with the assumption that the New Testament is the not just a collection of books but the church's Scripture. It is the Father's revelation of his Son given to his church through the power of the Holy Spirit.

This division of groups does not preclude conversations between them. In fact, clearer disciplinary boundaries will better allow both the history-of-religions approach and Scriptural theology to flourish as they seek truth together. Each will be done by people who

believe in the approach and endorse its methods. This sorting should not end dialogue across the groups but encourage it as each group presents its interpretations boldly and honestly. This dialogue should emphasize hermeneutics and methodology, places this essay could only lightly touch upon. The future of NTT belongs to approaches that seek truth; therefore, its future is one that first requires the self-sacrifice of descriptive approaches in order that something better may flourish.

**Funding:** This research received no external funding.

**Institutional Review Board Statement:** Not applicable.

**Informed Consent Statement:** Not applicable.

**Conflicts of Interest:** The author declares no conflict of interest.

## Notes

1. As to the relation of BT to NTT, Robert Morgan says that they are "closely related" (Morgan 1995, p. 104). Aligned with this, NTT can be thought of as a sub-discipline of BT that focuses on the New Testament. For example, see how Matera begins an article on NTT by looking at the origins of BT (Matera 2005, pp. 2–6).

2. Many books and articles have done this. The most helpful articles are (Matera 2005; Rowe 2006). An article that reviews many book-length contributions is (Schnabel 2019). Mead and Via have written books that give useful introductions to the field and its history (Mead 2007; Via 2002).

3. Heikki Räisänen and Thomas Hatina have written relatively recent books that give alternative views of what shape NTT should take (Hatina 2013; Räisänen 2000).

4. For example, Wrede says that NTT is to "lay out the history of early Chrisitan religion and theology" and that there is an "absolute necessity of going beyond the limits of the New Testament" when the "conception of the task" is clear (Wrede 1973, pp. 84, 101).

5. Hatina, for instance, classifies approaches according to a "foundationalist" or "dialectic" structuring (Hatina 2013, pp. 119–73). Mead classifies according to a work's issues, methods, and themes (Mead 2007).

6. Joel Green gives a helpful way of thinking about "history" when he describes three ways the term "historical criticism" is used in biblical studies. The first has as its goal the reconstruction of the past. The second excavates traditions in the text through traditional criticism, form criticism, source criticism, and redaction criticism. The third studies the historical context the biblical materials were written in (Green 2011, pp. 160–62). Here, the third use would be compatible with a textual focus whereas the first and second use with the history behind the text.

7. The purpose of giving specific examples is to make a fundamentally abstract and heuristic typology more concrete. The success of the typology does not require proper identification of examples and the reader should not get distracted by analyzing the placement of a particular NTT. No NTT will stick to one approach, for all mix history and textual interests, reality and description, neutrality and prescription, to various degrees. I explore the mixing of textual and historical interests in more detail elsewhere (Heringer 2014). A benefit of this typology is that it will encourage authors to think more clearly about the reasons behind such mixing.

8. This typology has structural parallels to the one Hans Frei created to explain biblical interpretation (Frei 1974, pp. 247–80). There, however, his typology examined where meaning resides in a text whereas this typology examines the subject matter of NTT. For more on Frie's typology see (Heringer 2018, pp. 43–53).

9. Timo Eskola uses historical background material and semiotics to investigate the metanarrative of the New Testament. For example, when discussing the resurrection, he remains descriptive in saying that the biblical accounts agree that a resurrection took place and leaves the ramifications of those claims to the words of the New Testament authors. As an example of mixing descriptive and reality approaches, however, he adds that his work supports the uncommon view that the historical Jesus anticipated his death and resurrection (Eskola, pp. 185–88).

10. Thomas R. Schreiner believes the Bible is the Word of God and thus makes true claims about reality and history (Schreiner 2008, pp. 886–88). This trust allows his focus to remain on describing the text without having to investigate its truth. Additionally, the assumption of truth shrinks the distance between the text and reader so that the mere description of the text feels prescriptive (see especially chp. 18). These assumptions mean the criticisms against descriptive approaches that arise later in this essay do not apply to Schreiner's work.

11. In this work, Frei distinguishes between "history" and "history-like" readings in order to argue that the meaning of the text lies in the narrative world it creates linguistically apart from its historical reference (Frei 1974, pp. 10–13, 280). Although Frei does not make a direct claim about the truth of this narrative world, his sympathetic description of premodern interpreters who believe that the world of the text is the real world points in this direction.

[12] Collins argues for "critical biblical theology" that clarifies "the meaning and truth claims" of ancient authors from a modern perspective. The neutral character of his approach is shown in that he believes the Bible cannot provide "objective, transcendent moral certainties", thereby stopping prescriptive readings of the text (Collins 2005, pp. 17–18, 78).

[13] Hatina serves as an example in the second and third stages of his approach where he locates NTT within religious studies. His approach is historical in its sociological study of the New Testament and descriptive in its "non-(a)theistic" methodology that does not "attempt to evaluate which claim is correct" among competing religions (Hatina 2013, p. 198). Prescription appears in the third stage where what is learned from a religious studies analysis of the New Testament is relevant to the modern world as it advocates for "universal human dignity, justice, and peace" (p. 215).

[14] The reality aspect of HRN and HRP needs further distinction from description. It is easy to see how historical claims can be either described or investigated, such as claims by a text, author, or community. The claims can be either left alone or investigated with an appropriate method. Yet, what is the truth value of a historical object that is part of the context surrounding the New Testament? For example, what is difference between a descriptive and reality approach to the temple? Since there is no claim being made aside from its mere existence, the distinction here is not obvious. My answer is that under the idea "temple" claims are being made, either by a variety of texts or archeology. A descriptive approach would describe these various claims; a reality approach would investigate those underlying claims to determine which were true.

[15] Craig L. Blomberg's work has a strong emphasis on the text but associates with the HR category because of traits such as a short defense of miracles in the gospels, an affirmation that Jesus felt abandoned on the cross, and an attestation of the historical reality of the resurrection (Blomberg 2018, pp. 71–72, 96–97). The main body of the text has a neutral feel because he moves most of his reflections on the modern relevance of the text to the concluding chapter (p. 15).

[16] Räisänen describes the two tasks of NTT as "the 'history of early Christian thought' (or theology, if you like), evolving in the context of early Judaism" and "critical philosophical, ethical and/or theological 'reflection on the New Testament', as well as on its influence on our history and its significance for contemporary life" (Räisänen 2000, p. 8). The first task sets out the HR characteristic; the second, P.

[17] Although it is often unrecognized, people evaluate every consequential claim they encounter. Any such claim is automatically run through plausibility considerations, such as the reliability of the speaker and comparison with what the person already knows to be true. In this manner, not every claim has to be relitigated anew. My argument is that we should not stop this process ad hoc but either admit the claims fails for some reason or continue this process to its end.

[18] All biblical quotations are from the CEB.

[19] Kavin Rowe has made a similar argument about the need to stop "deflecting" New Testament truth claims by examining a variety of texts that make claims that affect the reader (Rowe 2022, pp. 149–53).

[20] Morgan's work on NTT is respected enough to have merited a "festschrift" in his honor (Rowland and Tuckett 2006).

[21] All New Testament theologies have a section that discusses their methodology; however, most often these discussions remain short and deal with a whole range of topics from unity and diverity to hermenuetics. Broad discussions in a small space do not allow for the depth of engagement found in Morgan's two articles.

[22] Morgan, similar to many New Testament theologians, shows some ambivalence over whether he is interested in authorial intent or the grammatical meaning of the text. For example, he also says, "The exegete's contribution is to protect textual intention as the community attends to its Scriptures" (Morgan 2016, p. 386). This could be read as a grammatical and narrative interest in the text.

[23] A history-of-religions approach could take either an HDN or an HRP path. The difference between these two is whether the approach uses historical investigation to adjudicate the reality of the New Testament's claims.

[24] For example, Wrede says of NTT: ""We at least want to know what was believed, thought, taught, hoped, required, and striven for in the earliest period of Christianity; not what certain writings say about faith, doctrine, hope, etc." (Wrede 1973, p. 84).

[25] This statement is puzzling. A cursory study of biblical studies, or a special study of books on the historical Jesus, will show an intractable variety of opinions of who God and Jesus are. Is Jesus the Son of God or a man appointed by God to a special relationship with him? Was the cross the necessary step to Jesus's eventual triumph over death or the breaking of a man who threw himself against the wheel of history and was destroyed? The possible examples of incompatible visions of Jesus and God abound.

[26] Morgan's openness to "agnostic" theological interpreters is surprising as an agnostic interpreter would not share the mindset of the NT authors. I am unsure how he is able to maintain the distinction between his approach and Wrede's history-of-religions approach after making such an accommodation. I suspect his unwillingness to say that NTT must only be done by Christians is pushing him to make this pronouncement.

[27] This critique would be even stronger for an academically-acceptable HRP version of NTT. It would ask a confessional audience to accept a text that makes the stories and narrative of the Bible unrecognizable because its methodology would conclude that the stories were false. An HRP NTT author would say, this work is "theological" because I believe the same things you do (at some abstract or hypothetical level) even though the work tears down all the theological claims you hold dear.

[28] The differences between TRP and HRP are significant and worth exploring. A full discussion of these differences falls beyond the goals of this essay.

[29]     Young levels this critique at historical criticism itself: "I also suggest that spaces of the field attending primarily to description (traditionally: 'exegesis' or 'Historical Criticism') will be the most hospitable environments for mainstream protectionism. Fixating on description of New Testament texts can reproduce the idea of their obvious importance or centrality" (Young 2020, p. 339).

[30]     Elsewhere I argue that the current historical method forces such an approach because it is rooted in a misunderstanding of German historicism (Heringer 2018, pp. 1–41).

[31]     It is worth noting that this is opposite to the viewpoint of the history-of-religions approach. There, the person—her beliefs, character, and history—should not affect what is written. The methodological replaces the personal.

[32]     The TRN and HRN approaches are excluded from a group because I do not see a future for them in NTT.

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
