# Peer review of "Description or Truth? A Typology of New Testament Theology"

_religions, doi:10.3390/rel13060546_

Round 1
Reviewer 1 Report
I think this is a promising paper but it needs quite a bit more work before publication. Producing a typology is a good idea but at the moment I do not find the typology or the terminology clear, partly because I find the initialisms confusing (maybe that's just me). I wonder why you think there has to be one answer to the question that you pose (I don't think that all or maybe even most biblical theologians think that their way of doing it has to be the only way). And I'm puzzled at he way the paper moves rather suddenly from discussion of a typology to considering at length one approach (based on a couple or articles) that you don't favor before coming to your own suggestion. It seemed to make the paper disjointed. Within the typology section itself, I didn't recognize the names of a number of the scholars whose work is used as examples, but that may simply indicate my ignorance; on the other hand, there are are lots of better-known NTT scholars whom you don't mention. One little thing: is Frei really an example of TRP? Surely it's a purely historical study? (Would his book on the Identity of Jesus Christ be more relevant?) But the mention of this old book heightens the puzzle of why you mention some NTT works and not others.
Author Response
- Unclear typology (readability)
- I rewrote and rearranged parts of the typology section in order to help with clarity. Specifically, I set the three questions apart and moved them up in the section in order to let them better work with Figure 1. I also made minor changes throughout that section.
- Explain why I focus on Morgan and fix transition to him.
- I rewrote sections to focus on Morgan as an example of a descriptive approach to NTT. His example demonstrates that the the problems I identified with descriptive NTT have not been overcome in practice. Additionally, he shows that examples of descriptive NTT struggle to justify their "theological" name.
Reviewer 2 Report
Paper is interesting and considers an important issue. Morgan's interpretation seems to be an important aspect of the reflection on faith and religious experience. At the same time, it seems strong the awareness that the eschatological dimension of the New Testament must also be included in the perspective of historical development and change, which is why the content of the revelation must also be adapted to the historical awareness. The Paper seems to have grasped the problem well and points to a strong aspect of faith itself. The literature seems to be properly selected, although of course it is impossible to include everything and it is always a matter of the author's choice and preferences. It seems to me that the paper is suitable for publication and will be an interesting text for many readers who are interested in the issue. The advantage of the paper is that it seems easy to read also for readers less familiar with the issue.
Author Response
Please see the revised version
Round 2
Reviewer 1 Report
The article is much clearer than the first version. The transition to a detailed critique of an article by Morgan still seems odd. And it still does not persuade me that there has to be one answer to the author's main question. And I still cannot see that Frei fits in the category where the author puts him (and I am not competent to comment on many of the other examples as I have not read them).
